# Cryo-EM structure of a late pre-40S ribosomal subunit from *Saccharomyces cerevisiae*

André Heuer[1†]*, Emma Thomson[2†‡], Christian Schmidt[1], Otto Berninghausen[1], Thomas Becker[1], Ed Hurt[1,2]*, Roland Beckmann[1]*

[1]Gene Center Munich, Department of Biochemistry, University of Munich, Munich, Germany; [2]Heidelberg University Biochemistry Center, Heidelberg University, Heidelberg, Germany

**Abstract** Mechanistic understanding of eukaryotic ribosome formation requires a detailed structural knowledge of the numerous assembly intermediates, generated along a complex pathway. Here, we present the structure of a late pre-40S particle at 3.6 Å resolution, revealing in molecular detail how assembly factors regulate the timely folding of pre-18S rRNA. The structure shows that, rather than sterically blocking 40S translational active sites, the associated assembly factors Tsr1, Enp1, Rio2 and Pno1 collectively preclude their final maturation, thereby preventing untimely tRNA and mRNA binding and error prone translation. Moreover, the structure explains how Pno1 coordinates the 3'end cleavage of the 18S rRNA by Nob1 and how the late factor's removal in the cytoplasm ensures the structural integrity of the maturing 40S subunit.
DOI: https://doi.org/10.7554/eLife.30189.001

*For correspondence:
heuer@genzentrum.lmu.de (AH);
ed.hurt@bzh.uni-heidelberg.de (EH);
beckmann@genzentrum.lmu.de (RB)

[†]These authors contributed equally to this work

Present address: [‡]University of Sheffield, Sheffield, United Kingdom

Competing interests: The authors declare that no competing interests exist.

## Introduction

Ribosomes are the cellular machines that translate mRNAs into proteins. In eukaryotes, they consist of a small 40S and large 60S subunit, which carry the decoding and peptidyl transferase activity, respectively, and together comprise four ribosomal (r)RNAs (18S, 5.8S, 25S and 5S rRNA) and 78 ribosomal proteins in yeast. The synthesis of eukaryotic ribosomal subunits requires the concerted activity of ~200 assembly factors that drive ribosome biogenesis through a series of pre-rRNA cleavage, folding and modification reactions, which are coupled to the incorporation of ribosomal proteins (*Henras et al., 2015*; *Woolford and Baserga, 2013*; *Zemp and Kutay, 2007*). Initial steps of 40S biogenesis occur in the nucleolus, which leads to the formation of the first stable assembly intermediate, called the 90S pre-ribosome (*Dragon et al., 2002*; *Grandi et al., 2002*; *Kornprobst et al., 2016*), within which many of the early assembly steps for the 40S take place. This process requires between 50–70 different ribosome biogenesis factors (RBFs) (*Woolford and Baserga, 2013*; *Grandi et al., 2002*), which were shown by recent cryo-electron microscopy (cryo-EM) analysis to engulf the nascent pre-18S rRNA (*Kornprobst et al., 2016*; *Sun et al., 2017*; *Chaker-Margot et al., 2017*). Following early maturation steps, the pre-40S moiety detaches and is subsequently exported to the cytoplasm, containing only a handful of biogenesis factors including Pno1, Tsr1, Enp1, Ltv1, Nob1, Dim1 and Rio2 (*Schäfer et al., 2006*; *Schäfer et al., 2003*). Once in the cytoplasm, it has been proposed that assembly factors physically block the association of the translation machinery by occupying functional sites on the 40S subunit (*Strunk et al., 2011*). Structural insights into the architecture of pre-40S particles have previously been obtained through cryo-EM analysis, using preparations from both yeast and human cells (*Strunk et al., 2011*; *Johnson et al., 2017*; *Larburu et al., 2016*). In combination with RNA-protein crosslinking data, these structures have allowed the approximate positioning of most of the biogenesis factors on the late pre-40S particles (*Strunk et al.,*

2011; *Granneman et al., 2010*). However, in contrast to recent higher resolution structures obtained for the early 60S intermediates (*Greber, 2016*), no late pre-40S structures with atomic resolution are available. Accordingly, detailed insight into the molecular interactions of the RBFs and the conformation of the pre-rRNA in late 40S pre-ribosomes was lacking.

## Results and discussion

To gain a better understanding of the small ribosomal subunit biogenesis on a molecular level we purified late pre-40S particles via the well-defined biogenesis factor Ltv1 (*Schäfer et al., 2006*; *Johnson et al., 2017*), using Ltv1-Flag-TEV-ProteinA (FTpA) as bait (*Figure 1—figure supplement 1A*). With this strategy, we obtained a high yield of homogeneous pre-40S particles, which were used for single particle cryo-EM (*Figure 1—figure supplement 1B–C*). After classification we obtained a major class containing the stably bound RBFs Enp1, Tsr1, Rio2 and Pno1 (*Figure 1—figure supplement 1D*) but lacking a number of late binding ribosomal proteins (RACK1, uS10, uS14, eS10, eS26 and uS3) (*Ferreira-Cerca et al., 2007*). This main class could be refined to an average resolution of 3.6 Å, with the local resolution ranging from 3.5 Å in the core to approximately 8 Å for flexible regions (*Figure 1—figure supplement 2*). We built atomic models for Tsr1 and Pno1, and were able to model Enp1 and Rio2 on a secondary structure level (*Figure 1A* and *Figure 1—figure supplements 2–3*). In addition, the structure of the pre-18S rRNA revealed very distinct conformational differences, as compared to the mature state (*Heuer et al., 2017*), of functionally important regions including all three tRNA binding sites (A,P and E) and the entire mRNA path. We found that two major rRNA condensation steps still have to happen for these sites to mature: one in the head/beak region and the other in the central region of the 18S rRNA (*Figure 1B*).

The characteristic tertiary structure of the head rRNA (h28 to h43) is mainly determined by three-way junctions (*Mohan et al., 2014*). We observe that in the pre-18S rRNA only one such junction is not yet formed, namely that which connects h34, h35 and h38 (*Figure 1C* and *Figure 1—figure supplement 4A*). It joins three blocks of rRNA, which all contain parts of functionally important regions: one block contains h33 of the beak and h34, a central element in the formation of the A-site decoding center, while the block comprising h29-h32 and h38-h42 contains key residues for mRNA binding and accommodation of anticodon-arms for all three tRNAs. The third block (h35-h37) contains h36, which forms important tertiary interactions between head and body and is part of the central region of the 18S rRNA (*Wimberly et al., 2000*) (see below). Due to the absence of this junction, these blocks are shifted relative to each other and relative to the body, preventing the formation of the actives sites. Notably, formation of this junction requires the incorporation and stabilisation of the late associating ribosomal proteins uS3, uS10 and uS14 (*Lescoute and Westhof, 2006*), which are absent from the pre-40S particle (*Figure 2—figure supplement 1*).

The central region of the 18S rRNA comprises the central pseudoknot (CPK), a universally conserved structure that connects the head with the body via h28, h1 and h2. It provides a core structure around which major parts of the active A- and P-sites form, with the most central being h44 and h28. The tip of h44 contains two universally conserved adenosine bases (A1755/A1755, A1492/A1493 in *E. coli*) critical for mRNA decoding (*Ogle et al., 2003*) and the 'neck helix' h28 provides a hinge for head rotation, which is crucial for tRNA movement during elongation (*Mohan et al., 2014*; *Korostelev et al., 2008*). Formation of the CPK is a major structural landmark and we observe that, unlike in the 90S, the CPK is fully folded and the contact with the head (h36) has been established (*Figure 1—figure supplement 4B–C*). In contrast, we observe that the top of h44 is not yet base-paired, and the linker of h44 with h28 and h45 remains highly flexible. Notably, this linker forms major parts of the A and P sites in the mature state (*Figure 1D* and *Figure 1—figure supplement 4B*). Moreover, h44 is repositioned outwards relative to its mature position and h28 is tilted by 12 degrees in the direction of the beak (*Figure 1—figure supplement 4B–C*). Collectively, we observe that the pre-18S rRNA is still in a non-functional immature state since all elements forming the active decoding and mRNA interaction sites are prevented from adopting their mature fold (*Figure 2*).

We next investigated the role that the RBFs play regarding the immature pre-18S rRNA conformation. The first RBF, Tsr1, shares a similar domain architecture (I-IV) to several translational GTPases with an additional N-terminal extension, which was unresolved in previous studies (*Johnson et al., 2017*; *McCaughan et al., 2016*). Tsr1 mainly binds to the region which forms the universal translation factor binding site on the small subunit. Tsr1 contacts the junction of h5 - h15 and uS12 via its

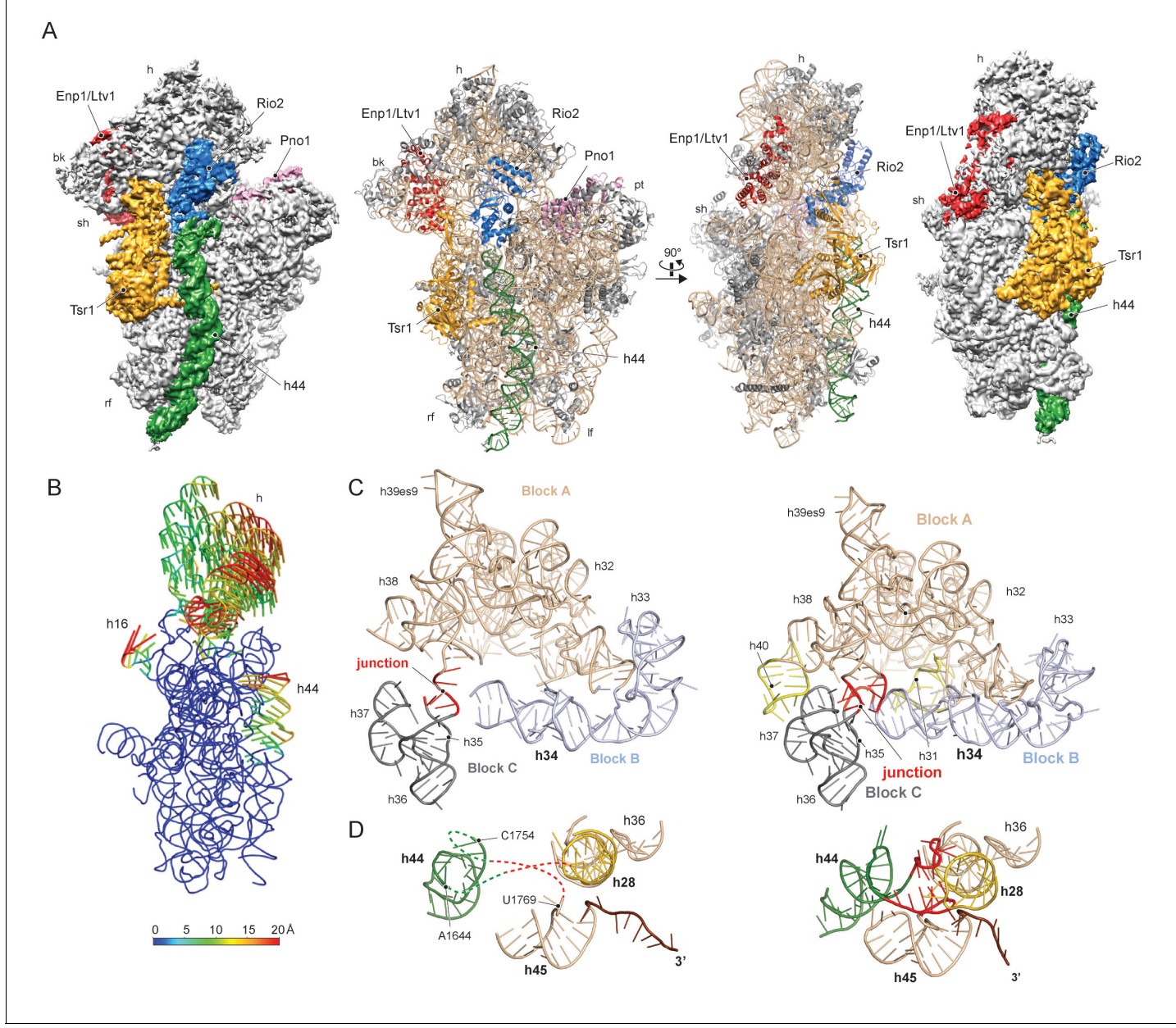

**Figure 1.** Structure of the pre-40S particle and conformation of the 18S rRNA.. (**A**) 3.6 Å cryo-EM reconstruction and molecular model of the pre-40S complex containing Enp1/Ltv1, Tsr1, Rio2 and Pno1 (**B**) Conformational transition of the 18S rRNA from the pre-mature to the mature state represented by vectors (superimposed on the 40S body). (**C**) Condensation of the head-forming rRNA on the h34/35/38 three-way junction from pre- (left) to mature state (right). Block A (h29–h32, h38–h42) mostly resembles the mature state and served as the moiety for superposition. Relative to block A, blocks B (h34 and 33) and C (h35–h37) are still shifted, since the three-way junction linking the blocks is not yet established. The connection of h40 to h37 and the loop of h31 are not established in the pre-state. (**D**) View focusing on the the linker between h44 with h28 and h45 from the central region of 18S rRNA. In the pre-state (left).he linker region as well as parts of h44 are unfolded and h28 (yellow) is tilted compared to the mature state (right).

DOI: https://doi.org/10.7554/eLife.30189.002

The following figure supplements are available for figure 1:

**Figure supplement 1.** Purification and cryo-EM of the pre-40S complex.
DOI: https://doi.org/10.7554/eLife.30189.003

**Figure supplement 2.** Assessment of resolution and model quality of the cryo-EM structure.
DOI: https://doi.org/10.7554/eLife.30189.004

**Figure supplement 3.** Fitting of the RBFs.
DOI: https://doi.org/10.7554/eLife.30189.005

*Figure 1 continued on next page*

Figure 1 continued

**Figure supplement 4.** Conformation and flexibility of the pre-18S rRNA

DOI: https://doi.org/10.7554/eLife.30189.006

domains I and III (*Figure 3A*). Specific interactions are established with h17 and to the N-terminal tail of eS30, which is yet to adopt its mature position (*Figure 4—figure supplement 1*). Further, domain IV of Tsr1 links the body with the immature head contacting h30 to h32, as well as the shifted h34. Notably, these rRNA structures are contacted from the opposite side by Enp1/Ltv1 (see below), resulting in the coordinated stabilization of the ribosomal beak in its immature conformation.

Due to high local resolution, we were able to build the N-terminal part of Tsr1. It forms a 35 Å long α-helix, which pierces through the ribosome between h5 and h44. By reaching further it touches h11-h12 (*Figure 3A–B*), thus serving as a distance enforcing wedge for h44. Thereby, via a long distance effect, Tsr1 keeps the linker connecting h44 with h28 and h45 unfolded and immaturely positioned. To assess the functional significance of the interaction of the N-terminal helix of Tsr1 and h44 we generated reverse charge mutants (R54D, K55D, K57D and K59D) where combinations of 3 or more substitutions indeed resulted in a slow growth phenotype (*Figure 3C*). All mutants showed the same nuclear localization, but a decrease in association with pre-ribosomes as compared to wild-type Tsr1 (wt) (*Figure 4—figure supplement 1A–B*). These data suggest that the N-terminus of Tsr1 is important for both, the stabilization of h44 in its immature conformation and the association of Tsr1 with pre-ribosomes. We also assessed the consequences of abolishing the interaction between domain IV of Tsr1 and the head of the pre-40S by removing domain IV. This mutant no longer supported yeast cell growth but continued to interact with pre-ribosomal particles (*Figure 4—figure supplement 1C–D*). We therefore propose that domain IV is not necessary for the association of Tsr1 with the pre-ribosome, but rather plays an important role to stabilize the pre-40S head in its immature conformation.

Enp1 is one of the few assembly factors that is already present in the 90S particle and remains associated until the integration of uS3 during late pre-40S biogenesis in the cytoplasm (*Kornprobst et al., 2016*; *Schäfer et al., 2003*). We observe Enp1 binding to the tip of the bent h16 near the mRNA entry site (*Figure 3D, 4A*). From there it reaches over to bind h32 - h34, thus keeping the head in its immature conformation together with Tsr1. Notably, we observe that Enp1 binds the same rRNA elements as in the 90S (*Kornprobst et al., 2016*; *Sun et al., 2017*; *Chaker-Margot et al., 2017*), further suggesting an early stabilization of the ribosomal beak. Ltv1, Enp1 and Rps3 are known to form a stable protein complex (*Schäfer et al., 2006*). Most likely due to the absence of Rps3 in our structure, only the extra density on top of Enp1 likely corresponds to its interaction partner Ltv1, which is in agreement with previous structural studies (*Strunk et al., 2011*; *Larburu et al., 2016*). The binding site of Enp1/Ltv1 occupies the position of the as yet unincorporated protein eS10 (*Figure 4A*) and explains Enp1/Ltv1's described role in facilitating uS3 integration at an adjacent site (*Schäfer et al., 2006*). It further suggests a role for Enp1 in the maturation of h34 and the h34-h35-h38 three-way junction.

Rio2 a RBF conserved in all archaea and eukaryotes (*Geerlings et al., 2003*; *Vanrobays et al., 2003*; *Schäfer et al., 2003*), is an essential serine kinase required for 40S maturation. It binds the pre-40S at the A and P site region with all three domains (*Figure 3E*). The N-terminal winged-helix-turn-helix motif (wHTH) contacts the tip of h18, which forms the 'latch' for the mRNA in the mature 40S together with uS3, h34 and uS12 (*Schluenzen et al., 2000*). The two-lobed kinase domains of Rio2, K1 and K2, are positioned close to h28 whereby K1 contacts the region, which serves as a hinge during the 40S head rotation (*Mohan et al., 2014*). Rio2's K1 also contacts the 40S head via h30 at a position close to domain IV of Tsr1. Finally, K2 contacts h29 and h42, which forms the P-site for binding the (initiation) tRNA in the mature ribosome.

We further identified Pno1, a factor that together with the endonuclease Nob1 controls one of the final events of 40S biogenesis, the maturation of the 18S rRNA through cleavage of the 3'-end at cleavage site D (*Lamanna and Karbstein, 2009*; *Lamanna and Karbstein, 2011*). This cleavage event is believed to be regulated by Pno1 (*Vanrobays et al., 2004*), which belongs to the family of single-stranded RNA binding proteins with KH-domains. It is located on the platform of the pre-40S (*Figure 5A*) where it interacts with uS11/uS1, the tilted rRNA h28, h45 and the 3'-end of the

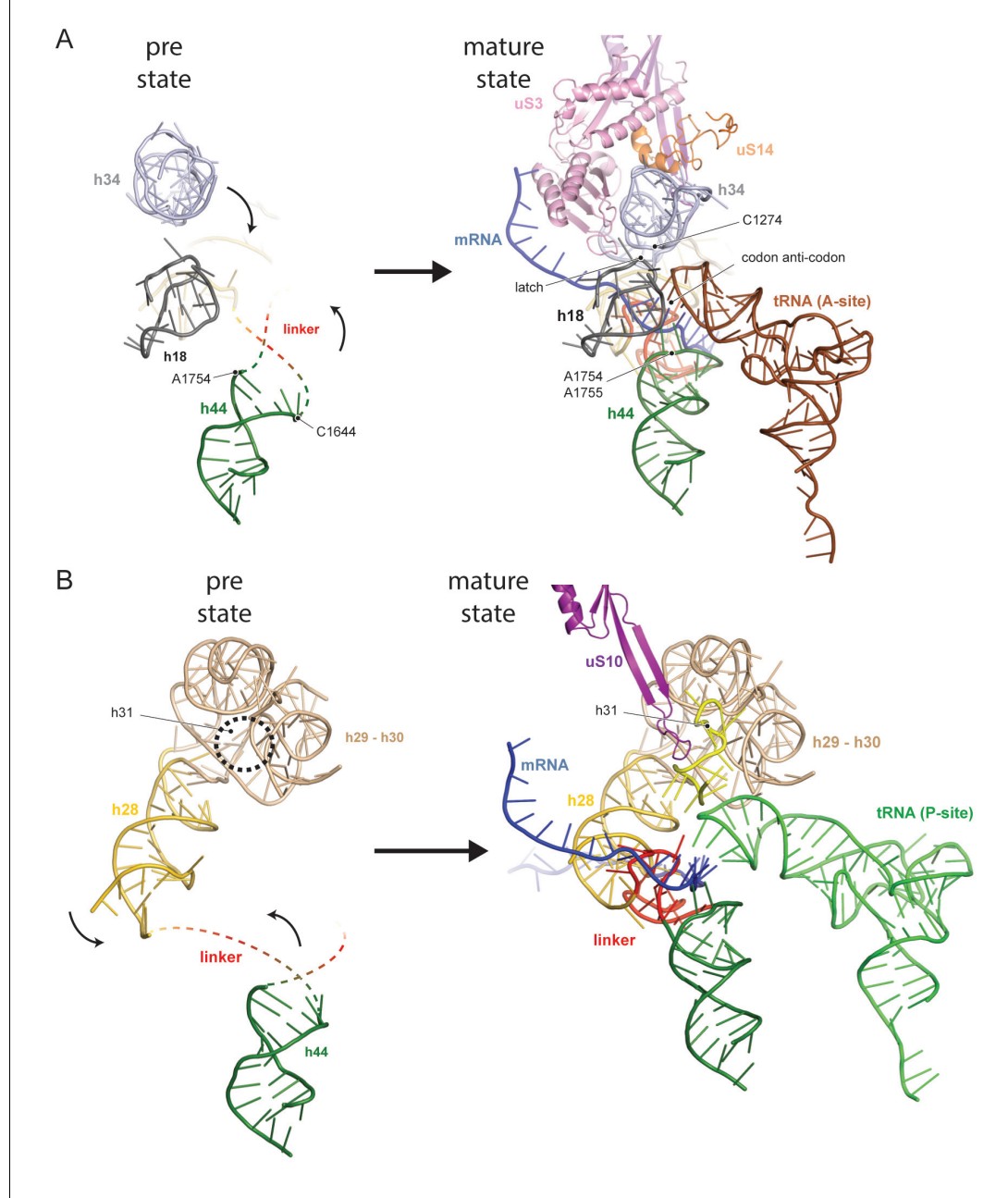

**Figure 2.** Comparison between pre- and native 40S focusing on the active sites. (**A**) View focusing on the mRNA entry and A-site. The A-site is composed of h18, h34 and h44, where in a translating ribosome the anticodon-loop of a A-tRNA is bound and the mRNA enters the 40S via the latch structure formed between h18, h34 and uS3. In the pre-40S, uS3 is absent and h34 is displaced. Moreover, h44 is shifted and its tip including the decoding adenines A1754 and A1755 is unfolded. (**B**) View focusing on the mRNA exit and the P-site. The P-site is composed of h24 (not shown), h28, h29 and h31 as well as the linker between h44 and h45. In the pre-40S this linker is delocalized and h28 is shifted. Moreover, the tip of h31 which is stabilized by uS10 in the mature state and binds the P-site tRNA in a translating ribosome is not folded in the pre-state.

DOI: https://doi.org/10.7554/eLife.30189.007

The following figure supplement is available for figure 2:

**Figure supplement 1.** Structure and environment of the native h34-h35-h38 three-way junction.

DOI: https://doi.org/10.7554/eLife.30189.008

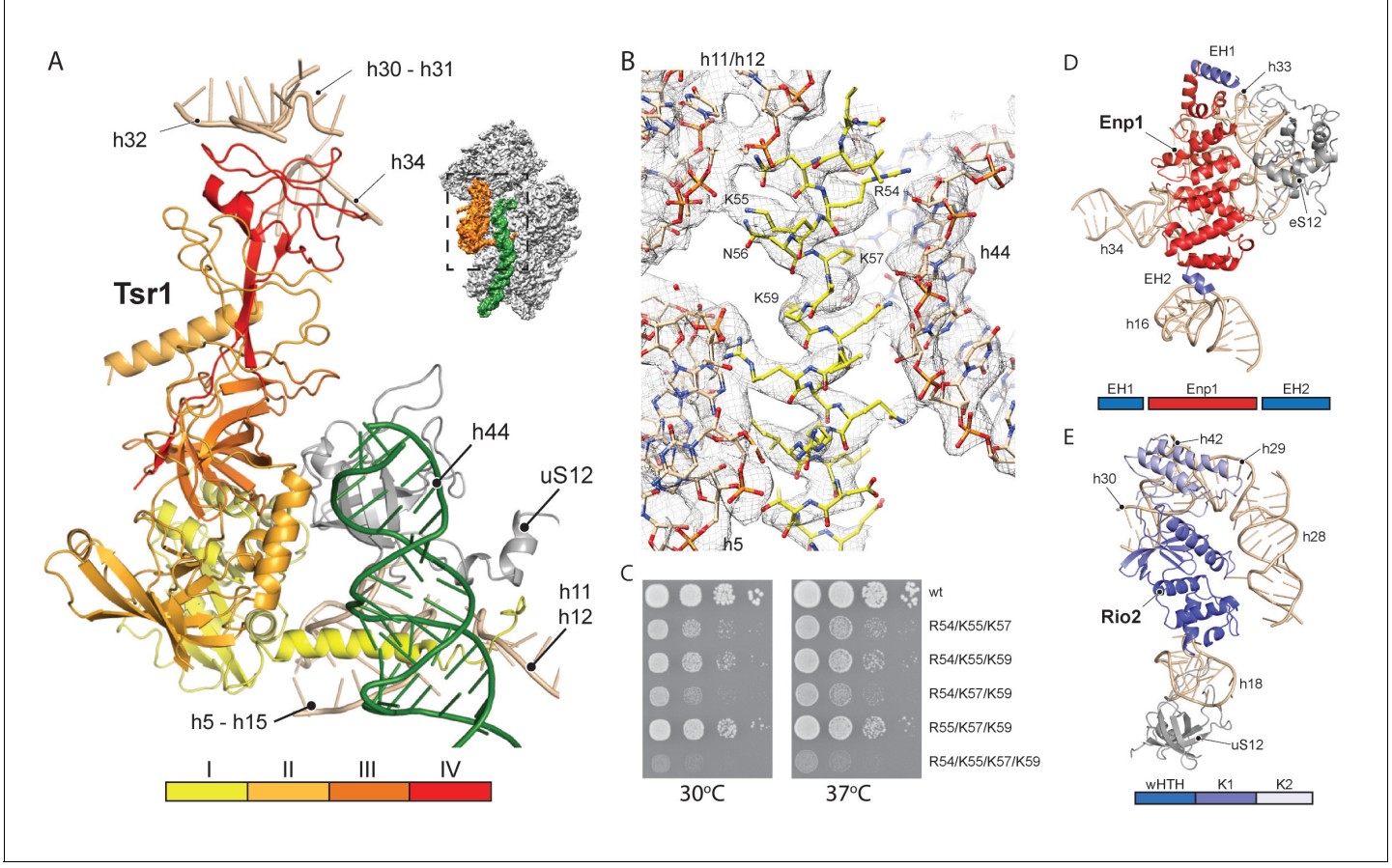

**Figure 3.** Interactions of Tsr1, Enp1/Ltv1 and Rio2 with the pre-40S. (**A**) Tsr1 binds the 40S body (h5–h15, h17 and uS12) via domains I and III, the 40S head (h30–h31, h32 and h34) via domain IV. The N-terminal α-helix of Tsr1 intercalates between h44, h5 and h11-h12. (**B**) Model of the Tsr1 N-terminal α-helixfitted into density low-pass filtered at 3.6 Å. (**C**) Growth analysis of wt Tsr1 and reverse-charge point mutations in residues interacting with h44 (R54D, K55D, K57D and K59D). Constructs were transformed into a Tsr1 shuffle strain and selected on SDC + FOA plates. Strains were spotted in 10-fold serial dilution on YPD plates and incubated for 2 days at the indicated temperatures. Different temperatures were used to assess if the growth defect observed at 30 degrees was enhanced at higher temperatures. (**D**) Enp1 binds to h33, h34 and eS12 and to the kinked h16; EH = extra helix for Enp1 or Ltv1. (**E**) Rio2 binds the 40S body via its N-terminal winged-helix-turn-helix-domain (wHTH) (h18, uS12), and the 40S head via the two-lobed kinase domain (K1/K2) (h28,h30). Moreover, K2 reaches into the P-site contacting h29, h42 and the region in between.

DOI: https://doi.org/10.7554/eLife.30189.009

The following figure supplement is available for figure 3:

**Figure supplement 1.** Functional analyses of mutant forms of Tsr1.
DOI: https://doi.org/10.7554/eLife.30189.010

pre-18S rRNA. Importantly, in this position, Pno1 sterically hinders h28 from adopting its mature conformation and the binding of eS26 (*Figure 4B*). We were able to follow the 3' rRNA end, bound by Pno1, up to the pre-terminal base (U1799, D cleavage occurs after A1800) in molecular detail: A multitude of interactions is formed by three α-helices of the KH2 domain of Pno1, which recognize the first two single-stranded bases (G1793 and A1794) as well as the stem of h45 (*Figure 5B*) including the last base of the h44-h45 linker (U1769), which later will form a part of the active P- and mRNA binding sites. Thus Pno1, like Tsr1 and Rio2, prevents compaction of the central region of 18S rRNA.

Like other members of the KH family (*Nicastro et al., 2015*), Pno1-KH2 uses its hallmark GXXG-RNA binding motif to position four nucleobases (residues 1795–1798) in a hydrophobic pocket (*Figure 5C–D*). Interestingly, KH1, which lacks the signature GXXG sequence in yeast (*Woolls et al., 2011*), also contributes to 3'-binding and contacts the terminal 18S rRNA bases 1797–1799 via its α-helix h1. Sequence alignments for the 3'-end revealed not only that the UCAU sequence -

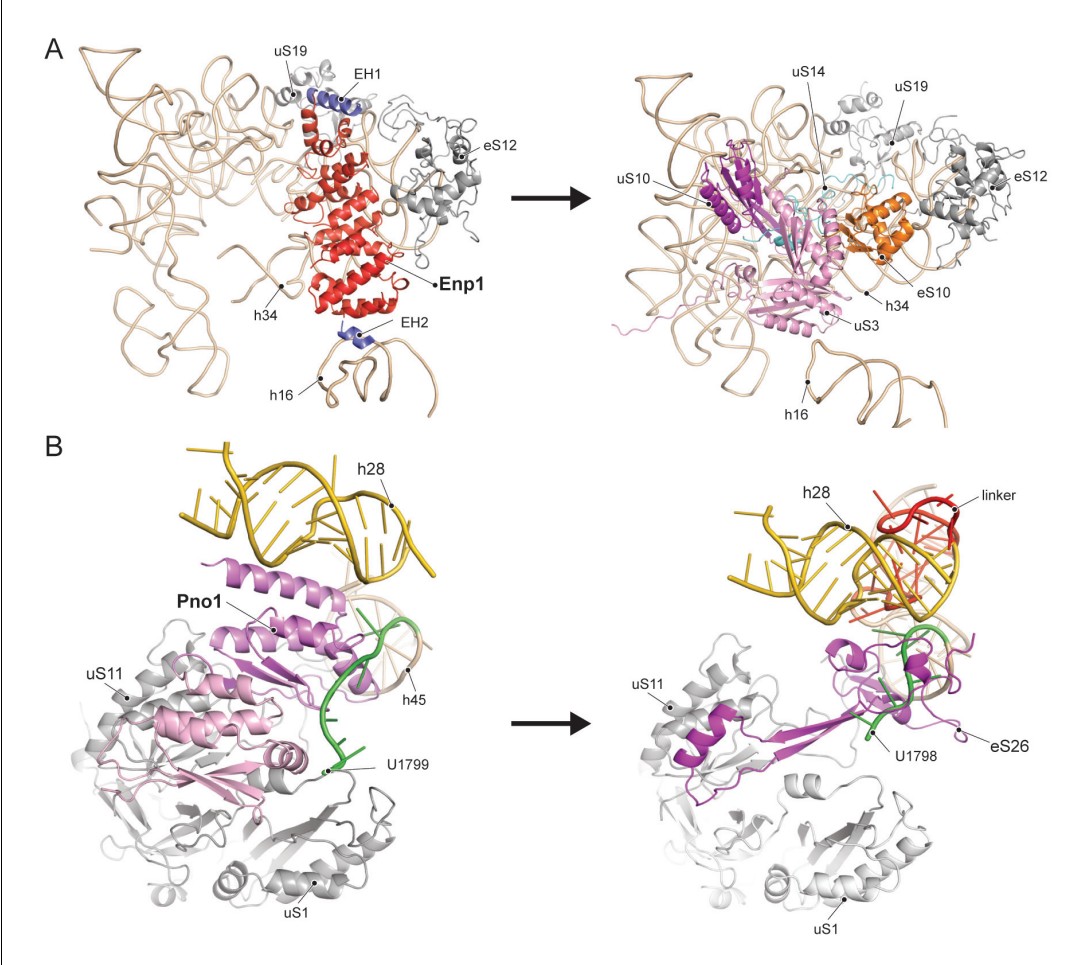

**Figure 4.** Comparison of RBF binding sites in the premature and the mature 40S. (**A**) Enp1/Ltv1 binds near the mRNA entry channel and connects the shoulder (h16) with the beak (h33, h34 and eS12). It occupies the position of the eS10. Moreover, uS3, uS10 and uS14 are not incorporated into the pre-40S particle. Note that h16 is in a bent conformation compared to the mature state. (**B**) Pno1 binds at the platform of the pre-40S contacting uS1, uS11, the kinked h28 as well as h45 and the 3'-end of 18S rRNA. It thereby occupies the position of eS26, which binds the rearranged 3' end in the mature state.

DOI: https://doi.org/10.7554/eLife.30189.011

The following figure supplement is available for figure 4:

**Figure supplement 1.** Binding of Tsr1 to h17 and eS30.

DOI: https://doi.org/10.7554/eLife.30189.012

that contactes KH2 - is conserved from yeast to humans, but also the surrounding bases up to the D-cleavage site. This suggests that specific binding of Pno1 and positioning of the 3'-end in a distinct conformation may be a universal feature of eukaryotic ribosome maturation. Notably, Pno1 is in an ideal location to sense any further maturation of 18S rRNA, in particular conformational changes of the close-by h28, which may allow Pno1 to productively present the D-site for cleavage by the neighbouring endonuclease Nob1. In addition, Pno1 may protect the 3' end against further cleavage until the small ribosomal subunit is fully matured.

In conclusion, we have discovered that the collective association of a few ribosome biogenesis factors on the late pre-40S ribosome regulates final rRNA folding steps at functionally important sites, in particular at the decoding centre. It appears that the role of these factors is to temporarily maintain the 40S subunit in a translationally incompetent state during ribosome biogenesis, preventing premature substrate interaction or entry into cycles of translation, which would be error-prone and potentially harmful to the cell. We envision that removal of biogenesis factors and the maturation of these regions are inter-dependent and coordinated processes. Conditional stepwise removal

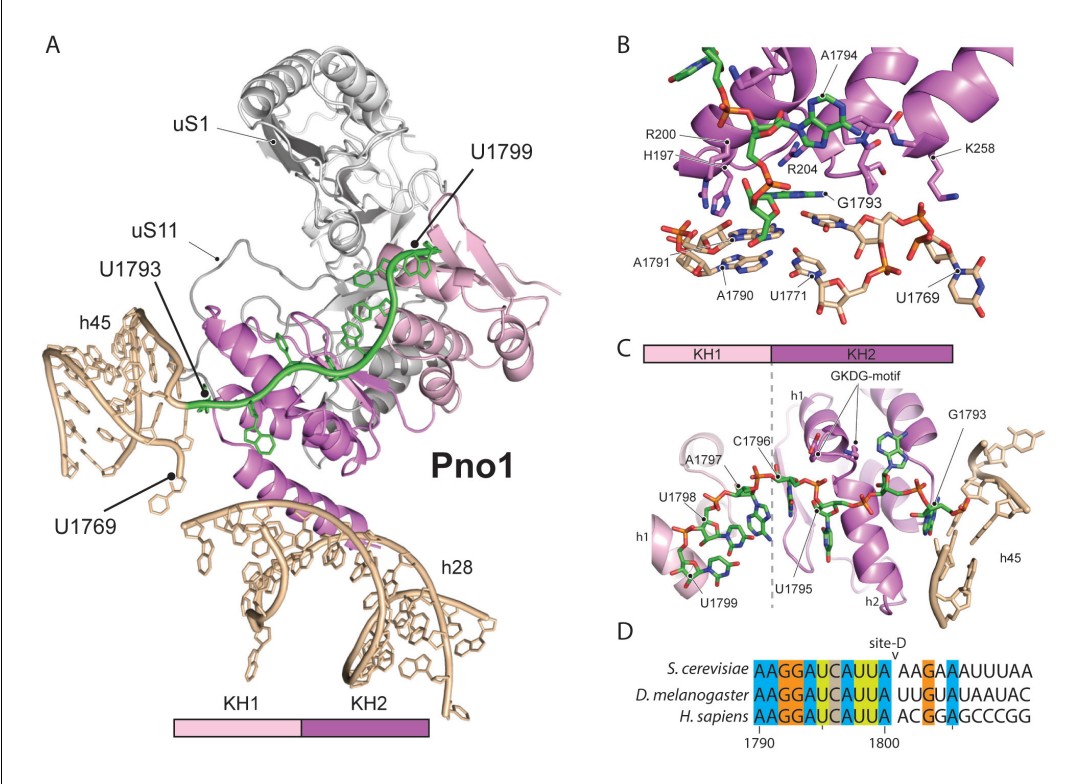

**Figure 5.** Molecular interactions of Pno1. (**A**) Pno1 binds uS11/uS1, h45, the tilted h28 and the 3' end (green) at the platform. (**B**) View focusing on the molecular interactions of KH2 with h45. Three C-terminal α-helices recognize the stem of h45 (A1791:U1770 and A1790:U1769), upon which G1793 is stacked. Specifically, Arg204 contacts G1793 and Arg200 together with His197 the backbone of the 3' strand (A1791 and A1790). The loop between two of the C-terminal helices of Pno1 (Gly253-Lys258) contacts U1770 and U1769 of the 5'-strand of h45. This loop also participates in binding of the first single-stranded base A1794 which is sandwiched between Pro256 and the GKDG-loop of KH2 (**C**) View focusing on the molecular interactions between Pno1 and the 3' rRNA end. Both KH domains interact with the 3'-end rRNA residues (G1793-U1799) which lead up to the D-cleavage site. The GKDG-loop of KH2 positions four nucleobases (U1795-U1798) close to its hydrophobic pocket and KH1 contacts the terminal bases (A1797-U1799) via h1 (**D**) Sequence alignment for eukaryotic 3' rRNA ends showing conservation up to the D-cleavage site.
DOI: https://doi.org/10.7554/eLife.30189.013

of these biogenesis factors may therefore serve as checkpoints that ensure the structural integrity of the ribosomal subunit, and thereby fitness for translation.

# Materials and methods

## Key resource table

| Reagent type (species) or resource | Designation | Source or reference | Identifiers | Additional information |
|---|---|---|---|---|
| strain, strain background (Saccharomyces cerevisiae) | Ltv1-FTpA;Tsr1shuffle | This paper | NA | Ltv1-FTpA- Genomic copy of Ltv1 tagged at the c-terminus with FLAG-TEV-proteinA tag; Tsr1 shuffle- genomic copy of Tsr1 deleted and rescued with a plasmid with a wild-type copy |

*Continued on next page*

*Continued*

| Reagent type (species) or resource | Designation | Source or reference | Identifiers | Additional information |
|---|---|---|---|---|
| genetic reagent (plasmids used for expression in Saccharomyces cerevisiae) | YCplac111-Tsr1-FTpA; YCplac111-Tsr1 R54D, K55D,K57D-FTpA; YCplac111-Tsr1 R54D, K55D,K59D-FTpA; YCplac111-Tsr1 R54D, K57D,K59D-FTpA; YCplac111-Tsr1 K55D, K57D,K59D-FTpA; YCplac111-Tsr1 R54D, K55D,K57D,K59D-FTpA; YCplac111-Tsr1-GFP; YCplac111-Tsr1 R54D, K55D,K57D,K59D-GFP; YCplac111-Tsr1DC86-FTpA | This paper | NA | Plasmids containing wild-type Tsr1 or the described point mutants. Expression is under the control of the Tsr1 promoter, and are tagged at the c-terminus with either FLAG-TEV-proteinA or GFP. |
| software, algorithm | EM-TOOLS | TVIPS GmbH | NA | http://www.tvips.com/ imagingsoftware/em-tools/ |
| software, algorithm | MotionCorr2.1 | https://doi.org/ 10.1038/nmeth.4193 | NA | http://cryoem.ucsf.edu/ software/driftcorr.html |
| software, algorithm | GCTF | https://doi.org/ 10.1016/j.jsb.2015.11.003 | NA | http://www.mrclmb. cam.ac.uk/kzhang |
| software, algorithm | Gautomatch | public | NA | http://www.mrclmb. cam.ac.uk/kzhang |
| software, algorithm | Relion-2 | https://doi.org/ 10.7554/eLife.18722 | NA | http://www2.mrclmb.cam.ac .uk/relion/index.php |
| software, algorithm | Phenix suite (phenix.real_space_ refine, molprobity) | Python-based Hierarchical ENvironment for Integrated Xtallography | RRID:SCR_014224 | https://www.phenix-online.org/ |
| software, algorithm | CCP4 (LIBG, ProSMART, Refmac5, COOT) | Collaborative Computational Project No. 4 Software for Macromolecular X-Ray Crystallography | RRID:SCR_007255 | http://www.ccp4.ac.uk/ |
| software, algorithm | UCSF Chimera | UCSF Resource for Biocomputing, Visualization, and Bioinformatics | RRID:SCR_004097 | http://www.cgl.ucsf.edu/chimera/ |
| software, algorithm | Pymol | PyMOL Molecular Graphics System, Schrödinger, LLC | RRID:SCR_000305 | https://pymol.org/ |

## Yeast strains and plasmids

For affinity purification of pre-40S particles for EM analysis, endogenous Ltv1 was tagged in a DS1-2b background (*Nissan et al., 2002*) at its C-terminus with a FLAG-Tev-protA (FTpA), as previously described (*Longtine et al., 1998*). All wt and mutant Tsr1-FLAG-Tev-protA or Tsr1-GFP constructs were expressed from plasmids under the control of the endogenous promoter. For Tsr1 affinity purification and localization studies, constructs were expressed in a BY4741 background. For growth analysis, constructs were transformed into a Tsr1 shuffle strain (in a BY4741 background), followed by selection on SDC + FOA. All constructs used in this study can be found in the key resource table.

## Affinity purification from yeast lysates

Purifications of all FTpA-tagged bait proteins were performed in buffer containing 50 mM Tris-HCl (pH 7.5), 100 mM NaCl, 1.5 mM $MgCl_2$, 5% glycerol, 0.1% NP-40, and 1 mM DTT. Cell lysates were prepared using a beadbeater (Fritsch), followed by centrifugation. Pre-equilibrated IgG Sepharose (GE) was added to the supernatant and incubated for 90 min at 4°C. Extensive washing was followed by TEV cleavage and a second step of purification on anti-FLAG MS2-agarose beads. Beads were washed, and proteins were eluted using a buffer containing 50 mM Tris-HCl (pH 7.5), 100 mM NaCl, 1.5 mM $MgCl_2$, 1 mM DTT, 3xFLAG peptide. For cryo-EM analysis the FLAG eluate was directly used. For Tsr1 purifications FLAG eluates were precipitated with TCA (10% final) and resuspended in

**Table 1.** EM data collection and refinement statistics.

| Data collection | |
|---|---|
| Particles | 84100 |
| Pixel size (Å) | 1.084 |
| Defocus range (μm) | 0.8-2.4 |
| Voltage (kV) | 300 |
| Electron dose (e$^-$ Å$^{-2}$) | 28 |
| **MODEL REFINEMENT** | Pre40S particle |
| **Model composition** | |
| Non-hydrogen atoms | 71923 |
| Protein residues | 4718 |
| RNA bases | 1635 |
| **Refinement** | |
| Resolution for refinement (Å) | 3.7 |
| Map sharpening B-factor (Å$^2$) | −92.8 |
| Average B-factor (Å$^2$) | 164.6 |
| FSC$_{average}$ | 0.85 |
| **R.m.s. deviations** | |
| Bond lengths (Å) | 0.0177 |
| Bond angles (°) | 1.61 |
| **VALIDATION and STATISTICS** | Pre40S particle |
| **Validation** | |
| Molprobity score | 2.20 |
| Clashscore, all atoms | 9.60 |
| Good rotamers (%) | 94.04 |
| **Ramachandran Plot** | |
| Favored (%) | 85.79 |
| Outliers (%) | 1.21 |
| **Validation (RNA)** | |
| Correct sugar puckers (%) | 97.8 |
| Good backbone conformations (%) | 65.6 |

DOI: https://doi.org/10.7554/eLife.30189.014

SDS sample buffer. Proteins were separated on 4–12% NuPAGE polyacrylamide gel and stained with colloidal Coomassie.

## Electron microscopy and image processing

Freshly prepared samples were adjusted to 1.5 A$_{260}$ (50 nM 40S ribosomes) and applied to Quantifoil R3/3 holey grids pre-coated with 2 nm carbon. Data was collected on a Titan Krios TEM (FEI Company) equipped with a Falcon II direct electron detector at 300 keV under low dose conditions of about 2.4 e-/Å2 per frame for 10 frames in total using the software EM-TOOLS (TVIPS) and a defocus range of −0.8 to −2.4 μm at a pixel size of 1.08 Å. Original image stacks were summed up and corrected for drift and beam-induced motion at micrograph level using MotionCor2 (*Zheng et al., 2017*). The contrast transfer function parameters and resolution range of each micrograph were estimated by GCTF (*Zhang, 2016*). All 2D and 3D classifications and refinements were performed with RELION-2 (*Kimanius et al., 2016*) after automated particle picking by Gautomatch (http://www.mrc-lmb.cam.ac.uk/kzhang/). Two-dimensional reference-free classification was performed to screen for particle quality (*Figure 1—figure supplement 1C*), non-ribosomal particles as well as poorly resolved 2D classes were discarded. 266.800 particles from good classes were

selected for 3D refinement using a mature 40S ribosome as reference. We performed two subsequent rounds of 3D classification in order to enrich pre-40S complexes (*Figure 1—figure supplement 1D*). First, the whole dataset was classified into 7 classes: class 1 and 2 contained orientation biased 40S ribosomes whereas classes 3 to 5 showed well-resolved 40S ribosomes with strong extra densities for the RBFs. In addition, class 6 showed poorly resolved pre-40S ribosomes and class 7 displayed a pre-40S ribosome with a very flexible head domain. The classes 3 to 5 were joined for movie refinement and a second round of 3D classification (six classes). Here, class 1 displayed distorted density due to orientation bias, while class 2 and 3 showed a very strong density for the majority of RBFs. Class 4 displayed a less distorted volume than class 1 but showed an extra density emanating from the platform to the head, which is likely Nob1 (*Strunk et al., 2011*). In addition Class 5 showed weak densities for RACK1, uS3 and the Dim1(*Johnson et al., 2017*). The most promising classes 2 and 3 were joined for final refinement and used for further interpretation. This final volume contained 84.100 particles was refined to 3.6 Å (FCS = 0.143) according to the 'gold standard' criterion, corrected for the modulation transfer function of the Falcon two detector and sharpened by applying a negative B-factor automatically estimated by RELION-2. Local resolution was calculated from 3.5 to 8.0 Å in steps of 0.5 Å using ResMap (*Kucukelbir et al., 2014*).

## Model building

For model building of the pre-40S subunit the structure of the mature *S. cerevisiae* 40S post splitting complex was used as a template (PDB 5LL6 [*Heuer et al., 2017*]). Available structures of the biogenesis factors Rio2 (PDB 4GYG [*Ferreira-Cerca et al., 2012*]), Tsr1 (PDB 5IW7 [*McCaughan et al., 2016*]), Pno1 (PDB 5WYJ [*Sun et al., 2017*]) and Enp1 (PDB 5WYJ [*Sun et al., 2017*]) were first fitted as rigid bodies into the isolated and appropriately low-pass filtered electron densities using UCSF Chimera (*Pettersen et al., 2004*). After rough docking, and manual adjustments where needed, flexible fitting and jiggle fitting was applied in Coot (*Brown et al., 2015*; *Emsley and Cowtan, 2004*). Regions not present in the available structures were modelled de novo, where the local resolution of the map allowed it (for example Tsr1, Pno1). rRNA which could, due to flexibility not be modelled with sufficient reliability, was removed from the model. In order to identify the extra density that may correspond to Ltv1, we fitted the structure of Enp1 found in the 90S pre-ribosome into the density. This fit left two rod-like densities unexplained, which we designated as extra helices 1 and 2 (EH1 and EH2). We speculated that these extra densities are either Ltv1 or an as yet unidentified part of Enp1. All models were subsequently combined and subjected to real-space refinement using PHENIX (*Adams et al., 2010*). After PHENIX refinement, the model was further subjected to reciprocal space refinement in REFMAC v5.8 (*Murshudov et al., 1997*) using restraints generated by ProSMART and LIBG as previously shown (*Brown et al., 2015*; *Amunts et al., 2014*). Because of the difference in local resolution and to avoid overfitting, h34, Enp1/Ltv1 and Rio2 were not subjected to REFMAC refinement. The final model was validated using MolProbity (*Chen et al., 2010*), summarized statistics are displayed in *Table 1*. Cross-validation against overfitting was performed as previously described (*Amunts et al., 2014*; *Fernández et al., 2014*) for both model refinements separately. Figures were created with the PyMOL Molecular Graphics System (Version 1.7.4, Schrödinger, LLC) and with UCSF Chimera.

## Accession codes

The EM density map is deposited in the 3D-EM database (EMD-3886) and the coordinates of the EM-based model is deposited in the Protein Data Bank (PDB-6EML).

## Acknowledgements

The authors thank S Lange, H Sieber, M Gnädig and S Griesel for technical assistance; M Thoms for the Ltv1-FTpA strain; L Kater for valuable assistance with high-performance computing; J Cheng for fruitful discussions. We thank the Leibniz-Rechenzentrum Munich (LRZ) for providing computational services and support.

## Additional information

### Funding

| Funder | Grant reference number | Author |
|---|---|---|
| Deutsche Forschungsgemeinschaft | FOR1805 | Roland Beckmann |
| Deutsche Forschungsgemeinschaft | GRK1721 | Roland Beckmann |
| Deutsche Forschungsgemeinschaft | SFB 646 | Thomas Becker<br>Roland Beckmann |
| Center for Integrated Protein Science Munich | CiPSM | Roland Beckmann |
| European Research Council | Advanced Grants CRYOTRANSLATION | Roland Beckmann |
| Deutsche Forschungsgemeinschaft | HU363/12-1 | Ed Hurt |
| Boehringer Ingelheim Stiftung | Boehringer Ingelheim Fonds PhD fellowships | Christian Schmidt |

The funders had no role in study design, data collection and interpretation, or the decision to submit the work for publication.

### Author contributions

André Heuer, Data curation, Formal analysis, Validation, Investigation, Visualization, Methodology, Writing—original draft, Writing—review and editing; Emma Thomson, Resources, Data curation, Validation, Writing—review and editing; Christian Schmidt, Validation, Writing—review and editing; Otto Berninghausen, Data curation; Thomas Becker, Conceptualization, Supervision, Writing—original draft, Writing—review and editing; Ed Hurt, Conceptualization, Funding acquisition, Writing—review and editing; Roland Beckmann, Conceptualization, Supervision, Funding acquisition, Validation, Writing—original draft, Project administration, Writing—review and editing

### Author ORCIDs

André Heuer (iD) https://orcid.org/0000-0001-6144-4316
Thomas Becker (iD) http://orcid.org/0000-0001-8458-2738
Roland Beckmann (iD) https://orcid.org/0000-0003-4291-3898

### Decision letter and Author response

Decision letter https://doi.org/10.7554/eLife.30189.016
Author response https://doi.org/10.7554/eLife.30189.017

## Additional files

### Supplementary files

• Transparent reporting form
DOI: https://doi.org/10.7554/eLife.30189.015

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
