## [Decision Letter]

Thank you for submitting your article "Atomic structure of a late pre-40S ribosomal subunit from *Saccharomyces cerevisiae*" for consideration by *eLife*. Your article has been favorably evaluated by James Manley (Senior Editor) and three reviewers, one of whom, Rachel Green (Reviewer #1), is a member of our Board of Reviewing Editors. The following individual involved in review of your submission has agreed to reveal their identity: David Tollervey (Reviewer #2).

The reviewers have discussed the reviews with one another and the Reviewing Editor has drafted this decision to help you prepare a revised submission.

Summary:

All three reviewers find the manuscript exciting and worthy of publication in *eLife*. And, of course, each reviewer has a few specific points that they wish to have addressed in a revised version of the manuscript (see below). Most of these points are for clarifications and some for figure revisions. There are no requests for additional experiments. The manuscript will not need to be sent out again. We thank you for submitting your excellent work to *eLife* and look forward to receiving a revised manuscript.

*Reviewer #1:*

This manuscript presents a cryoEM structure of a late pre-40S ribosome assembly particle isolated through pull-downs of a tagged well defined biogenesis factor Ltv1. The structure is at a much higher resolution than earlier studies and presents novel insights into the detailed structure of this assembly intermediate. Interestingly, the authors argue that instead of blocking the translational active sites, that the assembly factors hold key (rRNA) regions in inactive states such that final steps in maturation can occur. This is a somewhat different way of thinking about quality control during assembly. The manuscript was clearly written, nicely incorporating earlier observations, and identifying interesting features with mechanistic relevance. There are no equivalent structures in the field and this one should be of significant interest. I recommend publication.

*Reviewer #2:*

This paper reports a good structural analysis of a major, late precursor to the yeast 40S ribosomal subunit. The locations of the factors identified are in agreement with previous structural and biochemical data, but the molecular data provide fresh insights. The most significant biological finding is that the assembly factors generally hold the pre-rRNA in an immature conformation, rather than simply blocking access to functionally important sites as previously envisaged.

Several pre-ribosome structures have recently been published, but this is the first high resolution structure of a later pre-40S particle. The work is largely descriptive but is of high quality. Given the biological and medical importance of ribosome synthesis, the report is likely to be of wide interest.

1) A structure for Rio1 is shown in Figure 1—figure supplement 3, but it is not mentioned in the text. Where does this fit into the overall structure? Rio1 might not have been expected to be a stable or abundant component of these pre-40S particles and its apparent identification is interesting.

2) Nob1 is mentioned in the text as neighbouring Pno1, and would have been expected to be present in the pre-40S particle, but is not discussed. It is indicated in the purification (Figure 3—figure supplement 1). Nob1 and Rio1 are among the most interesting late pre-40S factors and some discussion is warranted.

*Reviewer #3:*

This work reveals the structures of biogenesis factors Tsr1, Enp1, Rio2 and Pno1 on a late 40S maturation intermediate. The proteins are resolved at varying resolutions, from near-atomic resolution allowing side-chain modeling, to lower-resolution sufficient to model secondary structures (for Enp1 and Rio2). This work is a substantial improvement over previous cryo-EM structures of the 40S subunit bound with these factors (e.g. ~20Å structures by Strunk et al., 2011).

The following changes are suggested prior to publication:

1) In the title, remove the word 'atomic', which is misleading as the readers would think the resolution of the structure is atomic. In the electron maps shown in figures, positions of atoms even in the most well-resolved regions (e.g. the long helix of Tsr1) are not unambiguously defined by the maps. The resolution of Enp1 is much worse than atomic.

Using a neutral term, such as "cryo-EM structure", is recommended.

2) In the Discussion of Tsr1, the authors say that the N-terminal domain "was absent in previous structures". Was the domain deleted from the protein or not modeled in the 3.6Å structure of Tsr1 that the authors refer to?

3) In the Discussion, the proteins are numbered – the first is Tsr1, the second is Enp1 etc. Does this reflect the binding order or other functional properties? This needs to be explained in the manuscript. If the numbering cannot be explained, it is confusing and should not be used.

4) Could Pno1 act to protect the 3' end against further cleavage, until the 40S is matured and the 3' end is protected by the 40S structure?

5) The sorting of cryo-EM data revealed a primary well-resolved class, which the authors have described. This is somewhat surprising given the abundance of biogenesis factors, and is likely the result of the purification strategy via Ltv1. Could the authors explain why this approach enriches for Tsr1, Enp1, Rio2 and Pno1, in the context of the 40S maturation mechanism?

---

## [Author Response]

Reviewer #2:[…] 1) A structure for Rio1 is shown in Figure 1—figure supplement 3, but it is not mentioned in the text. Where does this fit into the overall structure? Rio1 might not have been expected to be a stable or abundant component of these pre-40S particles and its apparent identification is interesting.

We apologize for this typing error. The model depicted is in fact of Rio2, not Rio1. This has now been amended in the figure legend.

2) Nob1 is mentioned in the text as neighbouring Pno1, and would have been expected to be present in the pre-40S particle, but is not discussed. It is indicated in the purification (Figure 3—figure supplement 1). Nob1 and Rio1 are among the most interesting late pre-40S factors and some discussion is warranted.

We could not gain any further information on the position of Nob1 from our cryo-EM data when compared to what is already known (Strunk et al., Science (2011) and Larburu et al., Nucleic Acids Res. (2016)). Since we lack Rio1 in our purification and thus in the structure, we cannot further elaborate on its relationship with Nob1.

Reviewer #3:[…] The following changes are suggested prior to publication:1) In the title, remove the word 'atomic', which is misleading as the readers would think the resolution of the structure is atomic. In the electron maps shown in figures, positions of atoms even in the most well-resolved regions (e.g. the long helix of Tsr1) are not unambiguously defined by the maps. The resolution of Enp1 is much worse than atomic.Using a neutral term, such as "cryo-EM structure", is recommended.

We agree with the referee and changed the title accordingly.

2) In the Discussion of Tsr1, the authors say that the N-terminal domain "was absent in previous structures". Was the domain deleted from the protein or not modeled in the 3.6Å structure of Tsr1 that the authors refer to?

The work referred to is the crystal structure of Tsr1 (U. M. McCaughan et al. (2016)), which was solved using an N-terminally truncated (Δ1-48) version of Tsr1. In this structure, in addition to the first 48 missing residues, also the N-terminal residues 49-66 could not be built, most likely due to disorder caused by the missing interactions with the ribosome. The phrasing in the manuscript has been adapted to reflect the comparison more clearly: “The RBF Tsr1, shares a similar domain architecture (I-IV) with several translational GTPases, however, it contains an additional N-terminal extension domain, which was absent or not structured in previous structural studies.”

3) In the Discussion, the proteins are numbered – the first is Tsr1, the second is Enp1 etc. Does this reflect the binding order or other functional properties? This needs to be explained in the manuscript. If the numbering cannot be explained, it is confusing and should not be used.

The numbering in the text did not relate to a specific binding or disassociation order. We agree with the referee that this could be confusing and adapted the text accordingly.

4) Could Pno1 act to protect the 3' end against further cleavage, until the 40S is matured and the 3' end is protected by the 40S structure?

We thank the referee for the suggestion. As also stated above, since we see that Pno1 binds to the rRNA at 3’ end almost until the D cleavage site, one can indeed speculate that Pno1 might protect the 3’ end against Nob1 cleavage until the small ribosomal subunit is fully matured. We added a sentence to the Discussion: “In addition, Pno1 might protect the 3’ end against further cleavage until the small ribosomal subunit is fully matured.”

5) The sorting of cryo-EM data revealed a primary well-resolved class, which the authors have described. This is somewhat surprising given the abundance of biogenesis factors, and is likely the result of the purification strategy via Ltv1. Could the authors explain why this approach enriches for Tsr1, Enp1, Rio2 and Pno1, in the context of the 40S maturation mechanism?

We agree with the referee’s notion that the stable association of our main pre-40S species with only the four factors Tsr1, Enp1, Rio2 and Pno1 is somewhat surprising. Unfortunately, we do not have a clear explanation why the purification strategy using tagged Ltv1 apparently enriches for this very stable and probably abundant intermediate of 40S maturation. However, the unusual stability/abundance of this complex is also illustrated by the fact that it has been described before, although at much lower resolution, for both, yeast and human (Strunk et al., Science (2011) and Larburu et al., Nucleic Acids Res. (2016)).